# Urobiome Signatures of Recurrent Urinary Tract Infections in Adolescent Pregnancy: A Longitudinal Study

**DOI:** 10.3390/microorganisms13102406

**Published:** 2025-10-21

**Authors:** Carlos Daniel Mora-Vargas, Orly Grobeisen-Duque, Oscar Villavicencio-Carrisoza, Diana Angélica Aguilar-Ayala, Orlando Castellanos-Diaz, Maria Guadalupe Martinez-Salazar, Alejandro Rosas-Balan, Veronica Flores-Rueda, Moises Leon-Juarez, Mario Guzman-Huerta, Lisbeth Camargo-Marin, Maria Isabel Villegas-Mota, Veronica Zaga-Clavellina, Ma. Guadalupe Aguilera-Arreola, Addy Cecilia Helguera-Repetto

**Affiliations:** 1Departamento de Inmunobioquímica, Instituto Nacional de Perinatología Isidro Espinosa de los Reyes, Ciudad de México 11000, Mexico; mora.vargas.daniel@hotmail.com (C.D.M.-V.); orly.grobeisen@gmail.com (O.G.-D.); cuauqbp@gmail.com (O.V.-C.); daguilar@unizar.es (D.A.A.-A.); salzarrg63@yahoo.com (M.G.M.-S.); moisesleoninper@gmail.com (M.L.-J.); v.zagaclavellina@gmail.com (V.Z.-C.); 2Escuela Nacional de Ciencias Biológicas del Instituto Politécnico Nacional, Ciudad de México 11350, Mexico; marreoag@ipn.mx; 3División de Ciencias Básicas e Ingeniería, Departamento de Ingeniería Eléctrica, Universidad Autónoma Metropolitana, Unidad Iztapalapa, Ciudad de México 09310, Mexico; ordcastellanos@gmail.com; 4Coordinación de Medicina de la Adolescente, Instituto Nacional de Perinatología Isidro Espinosa de los Reyes, Ciudad de México 11000, Mexico; alebalan1@yahoo.com.mx (A.R.-B.); veflorue@gmail.com (V.F.-R.); 5Departamento de Medicina Traslacional, Instituto Nacional de Perinatología Isidro Espinosa de los Reyes, Ciudad de México 11000, Mexico; mguzmanhuerta@yahoo.com.mx (M.G.-H.); lisbethcamargo@yahoo.com.mx (L.C.-M.); 6Secretaría de Salud del Estado de Quintana Roo, Chetumal 7700, Mexico; isabelvillegasmota@gmail.com

**Keywords:** adolescent pregnancy, urinary tract infection, UTI, recurrent UTI, microbiome, urobiome, UPEC, *E. coli*, Latin America

## Abstract

Adolescent pregnancy is a significant public health concern, with maternal and fetal risks compounded by pregnancy-related anatomical, hormonal, and urinary changes that predispose to urinary tract infections (UTIs). Alterations in the urinary microbiome may further influence infection susceptibility, yet little is known about its role during adolescent pregnancy. This study analyzed the urinary microbiome of adolescent pregnant patients and its association with UTI and recurrent UTI (rUTI) across gestation. Healthy adolescents were enrolled in the first trimester and followed through subsequent trimesters, with urine samples collected at each visit for microbiological diagnosis. Patients were classified as healthy (34 samples), single UTI (22 samples), or rUTI (31 samples), and oxford-nanopore 16S rRNA sequencing was used to assess taxonomic composition, microbial diversity, and operational taxonomic units. Distinct trimester-specific patterns were observed, with *Lactobacillus iners* progressively increasing and *L. kitasatonis* emerging as a dominant taxon during adolescent pregnancy. Interestingly, rUTI cases showed persistent *E. coli*, reduced *L. kitasatonis* and *L. ultunensis* in the second trimester, and the appearance of *Fannyhessea vaginae* (*Atopobium vaginae*) in the third. These findings suggest a potential microbial signature of rUTI in adolescent pregnancy, underscoring the need for personalized preventive strategies and the establishment of microbiome-based clinical cutoffs.

## 1. Introduction

Adolescent pregnancy represents a major public health concern that extends beyond its epidemiological burden. Its consequences are intertwined with physiological processes and have profound social implications, underscoring the need for an integrative understanding of its multifaceted impact. In 2023, 13% of girls under 18 years old gave birth, with 10% of these cases occurring in patients from South Asia, Latin American and the Caribbean [1]. In Mexico, the most recent published data reported that 18.5% of births were to adolescent mothers, representing a significant public health challenge due to the implications for both maternal and fetal health [2].

Due to the concomitant adolescent developmental stage and gestational state, maternal and fetal risks increase. Among the most frequent fetal outcomes are low birth weight (LVW), prematurity, stillbirth, extrauterine maladaptation, and congenital anomalies, largely due to inadequate prevention and insufficient prenatal care. These complications are commonly linked to the social and psychological burden adolescents face during this vulnerable period. Feelings of guilt, familial and social rejection, financial difficulties, lack of education leading to school dropout, and isolation from support networks further reduce the willingness of adolescents to seek medical care, often resulting in delayed recognition of pregnancy and missed opportunities for prevention [3].

Maternal adverse outcomes are often associated with the incomplete adaptation of immature physiological systems to the demands of pregnancy. These include preeclampsia, pregnancy-induced hypertension, preterm birth related to preterm premature rupture of membranes (PPROM), anemia, operative vaginal deliveries with forceps or vacuum, postpartum depression, and maternal mortality. Additionally, adolescents and pregnant women face an increased risk of developing urinary tract infections (UTI), which further exacerbate morbidity [3].

UTIs represent a significant health problem for women at all ages, particularly those of reproductive age. In Mexico, the second most commonly affected group is adolescents between 15–19 years old. Pregnancy-related anatomical and hormonal changes, including ureteral dilation due to increases in progesterone and relaxin, changes in vesicoureteral reflux, urinary stasis, and chemical alterations in the urine that make it more alkaline, further predispose to UTI development. Moreover, changes in the microbiome may influence the acquisition and progression of UTIs, most frequently caused by *Escherichia coli*, the predominant pathogen in women [4,5].

In the last decade, thanks to new sequencing techniques, the presence of unique bacterial communities inhabiting the urinary tract in a non-pathogenic manner has been described. Ongoing research has aimed to characterize the genetic and metabolic composition of these communities [6], broadening the perspective on the urinary microbiome or urobiome, and marking the end of the paradigm that dictated that “healthy urine is sterile” [7]. This shift in perspective has led to new questions, such as: What is the composition of the female urinary microbiome (FUM)? What factors influence its composition? And what role does the FUM play in women’s health?

To date, we know that the healthy FUM is predominantly composed of species from the phyla *Firmicutes*, followed by *Bacteroidetes*, *Actinobacteria*, and *Proteobacteria* [8,9,10,11]. Studies analyzing urine samples from healthy adult women found that nearly 88% of microbial abundance was attributed to *Lactobacillus*, *Prevotella*, and *Gardnerella* [11], with lesser representation of organisms belonging to the genera *Streptococcus*, *Staphylococcus*, *Prevotella*, *Anaerococcus*, *Corynebacterium*, *Fannyhessea* (*Atopobium*), *Bifidobacterium*, *Enterococcus*, *Escherichia*, *Actinobaculum*, *Aerococcus*, *Peptoniphilus*, *Sneathia*, *Veillonella*, *Shigella*, *Actinomyces*, *Allisonella*, *Alloscardovia*, *Anoxybacillus*, *Arthrobacter*, *Burkholderia*, *Dialister*, *Finegoldia*, *Klebsiella*, *Ralstonia*, *Rhodanobacter*, and *Trueperella* [8,9,11,12]. Nevertheless, during female growth and development, the FUM undergoes marked transformations influenced by hormonal, physiological, and anatomical changes. Both the urinary and reproductive tract microbiomes evolve throughout life, with age-dependent shifts in diversity and composition. There is evidence showing that prepubertal girls harbor a more diverse microbial community, with higher abundances of *Prevotella*, *Dialister*, and *Campylobacter*, whereas adolescents show a transition toward a *Lactobacillus*-dominated environment in the urethra and vagina, mainly represented by *L. gasseri*, *L. iners*, and *L. jensenii*. As women reach adulthood, the urinary microbiome becomes more stable and enriched in *Lactobacillus* species, particularly *L. crispatus*, which plays a key protective role in maintaining urogenital homeostasis [5]. Additionally, it is well recognized that both infection and antibiotic treatment can alter the urinary microbiome, yet little is known about microbial community changes during and after infection [13,14]. Various studies have identified differences in species abundance between UTI patients and healthy controls, highlighting the presence of *K. pneumoniae*, *Streptococcus agalactiae*, *Aerococcus urinae*, *E. faecalis*, *E. coli*, *Staphylococcus aureus*, and *Streptococcus anginosus* in UTI patients [15]. Other studies have detected polymicrobial communities and antimicrobial resistance genes in urine samples [16]. Additionally, three microbial signatures have been identified in urine: two not associated with UTIs, dominated by *Actinobacteria* and *Firmicutes*, and one associated with UTIs, characterized by *Proteobacteria* and common pathogens such as uropathogenic *E. coli* (UPEC), *Klebsiella*, *Pseudomonas*, and *Enterobacter*. Newly identified UTI-associated bacteria include *Acidovorax*, *Rhodanobacter*, and *Oligella*, as well as viruses and bacteriophages, highlighting the need for further studies on the viral component [17].

To date, there are only a few metagenomic studies of UTI and recurrent UTI patients, and longitudinal studies are needed to better understand the microbial ecology and pathobiology of these infections. Additionally, due to the non-specificity of the populations included in each study, variables such as age, race, geographic distribution, and pregnancy status remain poorly defined. This makes it difficult to determine whether the described communities are specific to a social group or influenced by other conditions. Therefore, our study aims to analyze the urobiome in adolescent pregnant patients and further describe its correlation with UTI and recurrent UTI (rUTI) during each trimester.

## 2. Results

The 86 urine samples from 45 patients were divided as follows: 34 samples corresponded to control patients, 22 samples to UTI patients and 31 samples to rUTI patients. Samples were also divided by trimester of gestation.

A total of 2,281,869 sequences were obtained after quality processing, with an average of 27,944 reads per sample. A total of 96 genera and 224 species were identified. To simplify visualization and focus on the most relevant taxa, only the 15 most abundant genera were included in the analysis; the remaining genera were grouped under the category “Other.”

Relative abundance analysis of the 15 most abundant genera was performed across the different clinical groups (Control, UTI, and rUTI) and trimesters of pregnancy (Figure 1). *Lactobacillus* was the most abundant genus overall, particularly in second and third trimester samples, regardless of clinical group. In contrast, samples from the first trimester (especially in the UTI and rUTI groups) showed a more heterogeneous microbial composition, with higher relative abundances of *Escherichia*/*Shigella*, *Gardnerella*, *Prevotella*, *Streptococcus*, and other genera. Notably, *Escherichia*/*Shigella* was markedly more abundant in UTI samples from the first trimester, suggesting its potential role in early gestational urinary tract infections. Across all trimesters, control samples tended to show higher proportions of *Lactobacillus*, indicating a more stable and typical vaginal microbiota. These patterns suggest dynamic microbial shifts throughout pregnancy that may be influenced by both trimester and infection status.

To focus the analysis on the most relevant taxa, the 224 species found were filtered according to criteria proposed by Fettweis: species were retained if (1) they reached ≥1% relative abundance in at least 5% of samples, or (2) ≥0.1% in at least 15% of samples [18]. After filtering, 23 species were included in downstream analyses.

Analysis of the relative abundances by gestational trimester and sample type (Control, UTI, and rUTI) revealed dynamic shifts in microbial composition (Figure 2). In the first trimester, control samples were dominated by *Lactobacillus kitasatonis* (39.7%), *L. iners* (16.5%), and *Megasphaera sueciensis* (9.9%), with lower abundances of *Gardnerella vaginalis*, *Fannyhessea vaginae* (*Atopobium vaginae*), *Prevotella timonensis* and *L. acidophilus* and *L. helveticus*. UTI samples showed dominant abundance of *Escherichia coli* (38.8%; not present in control patients), *L. kitasatonis* (24%) and *M. sueciensis* (14.5%), and a lower abundance of *L. iners* with respect to the control samples was quantified (4.3%). rUTI samples were predominantly colonized by *E. coli* (48.9%), followed by *L. iners* (19.5%) and *L. kitasatonis* (18%). Notably, *E. coli* abundance was significantly higher in rUTI versus controls (*p* = 0.05; Appendix A).

During the second trimester, control samples exhibited a microbiota dominated by *L. kitasatonis* (50%) and *L. iners* (25.5%). UTI samples were dominated by *L. kitasatonis* (41.1%), *L. prophage* (13.2%), and *L. helveticus* (7.7%); interestingly, *E. coli* disappeared. In contrast, rUTI samples retained dominance of *E. coli* (40%) and showed the appearance of *Streptococcus dentisani* (12.8%). These compositional changes showed that UTIs during this period change the typical abundance of Lactobacillus species observed in control patients, with an important disruption of their dominance in rUTI patients (37.3% of *Lactobacillus* spp. presence in rUTI versus 83.6% and 93.5% in control and UTI patients, respectively). A significant difference in the abundance of *L. kitasatonis*, *L. ultunensis* and *E. coli* was observed between control and rUTI samples (*p* ≤ 0.05; Appendix A).

In the third trimester, all groups showed *Lactobacillus* dominance, although with varying species composition. Control samples were primarily composed of *L. kitasatonis* (51%) and *L. iners* (30%). UTI samples were dominated by *L. kitasatonis* (49%), *L. iners* (15.2%) and other Lactobacillus species including *L. helveticus* and *L. fornicalis*. rUTI samples presented a more heterogeneous profile, with *L. kitasatonis* (41.2%), *L. iners* (16%), and persistent *E. coli* (11%). Importantly, *Fannyhessea vaginae (A. vaginae)* was detected exclusively in rUTI samples (Appendix A).

Overall, the data demonstrate trimester-dependent shifts in microbial composition and suggest that *E. coli* presence, particularly in recurrent infections, may impair the establishment of a healthy *Lactobacillus*-dominated community.

Analysis of alpha diversity metrics (Figure 3) revealed that, during the first trimester, *evenness* and Shannon indices were similar across the three cohorts (Figure 3A,B). However, as pregnancy progressed, alpha diversity (evenness and Shannon indices; Figure 3A,B) decreased in control patients while increasing in those with UTI and rUTI. This trend is consistent with the species composition observed in each cohort: during the first trimester, control samples displayed a higher number of species compared to UTI and rUTI samples, but this relationship was reversed in the later trimesters (Figure 3C).

These findings align with previous results showing that dominance by *Lactobacillus* species (particularly in control samples) may suppress the relative abundance of other taxa, contributing to lower alpha diversity [5,19].

Beta diversity was assessed using multiple indices (Canberra, Jaccard, and Bray–Curtis), revealing a distinct cluster that included only a subset of UTI and rUTI patients (Appendix A). The absence of statistically significant differences among groups suggests that the overall microbial community composition remains largely similar, likely reflecting the presence of a shared core microbiota. Consequently, the variations observed in alpha diversity are more indicative of changes in evenness or relative taxa abundance rather than a complete species turnover between groups. This pattern is characteristic of a relatively stable microbial system, where functional or abundance-related alterations occur without major shifts in overall community structure [20,21], yet still signify a dysbiotic state.

The last results are consistent with the observed urotype distribution (Figure 4), where control samples showed Lactobacillus-dominated profiles, while rUTI samples frequently exhibited *E. coli*-dominated urotypes. In fact, the *E. coli*-dominated urotype was observed exclusively in samples from patients with single or recurrent UTIs, and only during the first trimester, where it was the most abundant. In the second trimester, this urotype persisted only among patients with rUTIs. Finally, by the third trimester, its abundance decreased substantially, being replaced by indeterminate urotypes with no clear dominant genus or species. Urotype classification revealed a clear relationship between *Escherichia coli* dominance and urinary tract infection status, specifically in rUTI along pregnancy.

Fisher’s exact test was used to assess the relationship between urotype distribution and clinical or gestational variables. No significant association was found between urotype distribution and gestational trimester across the three patient groups (*p* ≥ 0.05). However, when comparing patient groups regardless of trimester, urotype distribution was significantly heterogeneous (*p* = 0.007364). These findings suggest that infection status, rather than pregnancy stage, plays a more critical role in shaping urinary microbiota composition as defined by urotype.

Finally, we analyzed how the abundance of each microorganism varied throughout gestation across our three cohorts (Figure 5). Interestingly, *L. iners* showed a gradual increase over the course of pregnancy in the control group, whereas in patients with UTI, its abundance peaked during the second trimester and declined in the third. Strikingly, in the rUTI cohort, the abundance of *L. iners* remained consistently moderate across all three trimesters, showing a small decrease during the second trimester. Another noteworthy microorganism is *L. kitasatonis*, which displayed elevated levels across all trimesters. In the control group, its abundance was high in the first trimester and progressively increased thereafter. A similar trend was observed in the UTI cohort, whereas in the rUTI cohort its abundance remained low with a relative increase during third trimester.

Two other species with low abundance, *L. helveticus* and *L. acidophilus,* showed interesting differences along pregnancy. In control patients, their abundances were low but noticeable during the first and second trimesters, followed by a marked decline in the third trimester. In contrast, in both the UTI and rUTI cohorts, these species exhibited low abundance in the first trimester but progressively increased throughout pregnancy, reaching their highest levels in the third trimester.

As expected by our previous results, *E. coli* showed a pronounced peak during the first trimester in the UTI cohort. Among rUTI patients, its abundance was high in the first trimester, followed by a slight decrease from the second trimester onward, persisting at lower levels into the third trimester. In contrast, *E. coli* was not detected in control patients.

## 3. Discussion

In this study, we characterized the dynamics of the urinary microbiome during pregnancy in three groups of Mexican adolescent women: healthy controls, patients with urinary tract infection (UTI), and those with recurrent UTIs (rUTI). Our findings revealed differential patterns in the abundance of *Lactobacillus* spp. across trimesters, suggesting that pregnancy modulates microbial composition in distinct ways depending on clinical condition.

A relevant finding was the progressive increase of *L. iners* and *L. kitasatonis* in control women as pregnancy advanced. This is particularly interesting given that *L. iners* has been described as an ambiguous species, due to the fact that it has been found in both healthy women and dysbiotic states, and has been proposed as a “transitional taxon” between eubiosis and pathological states [22]. *L. iners* displays unique genomic and physiological features, such as possessing the smallest genome among *Lactobacillus* spp. (~1.3 Mb), which implies the loss of several metabolic pathways [23]. For instance, in terms of lactic acid metabolism, *L. iners* can only produce the L-lactic acid isomer, unlike *L. crispatus*, *L. gasseri*, and *L. jensenii*, which produce both L- and D-lactic acid [23]. Production of D-lactic acid has been associated with stronger antimicrobial activity and protective immunomodulatory effects, explaining why microbiomes dominated by *L. crispatus* are linked to a lower risk of dysbiosis, whereas *L. iners* dominance results in a less acidic pH and lower protective capacity [24]. Furthermore, *L. iners* lacks the enzymatic machinery to produce hydrogen peroxide (H_2_O_2_), further reducing its defensive potential against pathogens [22].

Nonetheless, *L. iners* exhibits strong adhesion capacity to host epithelia by encoding a fibronectin-binding protein with a motif similar to FnBPA of *Staphylococcus aureus* [25]. McMillan et al. demonstrated that *L. iners* binds significantly more strongly to human fibronectin than other *Lactobacillus* spp. at near-neutral pH, which may favor its persistence despite reduced acidification [25]. Since fibronectins are naturally produced by the urothelium [26], it has been suggested that, although *L. iners* is not the most efficient lactic acid producer, in neutral pH conditions its high biomass may act as a physical barrier against pathogen proliferation, partially compensating for its metabolic limitations. This is the first study to identify *L. iners* as a prominent member of the urinary microbiome in Mexican adolescent pregnant women.

Also, a study in Chinese healthy pregnant women revealed that *L. iners* dominated vaginal communities displayed enrichment in genes related to folate biosynthesis, glycosyltransferases, and antimicrobial resistance (e.g., *erm*B), suggesting functional versatility and an adaptive capacity to the changing hormonal and immune landscape of pregnancy [27].

These findings highlight the potential role of *L. iners* not only as a transitional taxon but also as a key player in the dynamic adaptation of the urinary microbiome in adolescent pregnancies, a group that has been underrepresented in previous studies.

In our UTI patients, *L. iners* abundance increased from the first to the second trimester but declined in the third, possibly reflecting a loss of urinary ecosystem stability under infectious stress. In rUTI patients, it displayed the opposite trend to controls, progressively decreasing throughout pregnancy. These results reinforce the view that *L. iners* protects by acting as a physical barrier against other microorganisms establishment, then in rUTI patients urobiome is more diverse.

We also identified differential patterns in *L. acidophilus* and *L. helveticus*. In control women, the abundance of both species decreased as pregnancy progressed, whereas in UTI and rUTI they showed the opposite pattern. Previous studies of the vaginal microbiome have shown that, in healthy pregnancies, the microbiota tends to stabilize and reduce its diversity [5,19]. Besides diversity, species abundance also changes in pregnancy, as observed with the decrease in *L. helveticus* [28], while species such as *L. crispatus* or *L. gasseri* predominate [29]. The expansion of *L. acidophilus* and *L. helveticus* that we observed in infected patients could represent an adaptive response of the urinary ecosystem to the imbalance caused by pathogenic bacterial colonization. Such increase may reflect a compensatory, potentially protective role, whereby the microbiota “attempts” to mitigate dysbiotic effects.

Another novel observation was the behavior of *L. kitasatonis*: its abundance increased gradually in both controls and UTI patients, whereas in rUTI it showed a fluctuating pattern (high in the first trimester, low in the second trimester, and returning to higher levels in the third trimester). To the best of our knowledge, this is the first report documenting the presence of *L. kitasatonis* in the human urinary microbiome as a dominant microorganism, as no previous studies have described it in this anatomical site. *Lactobacillus kitasatonis* is known to produce both L- and D-lactic acid isomers, thereby contributing to the acidification of its environment [30]. Additionally, this species carries a homolog of the helveticin gene from *L. helveticus*, an antibacterial peptide (bacteriocin) that disrupts bacterial cell membranes and walls [31]. The function of *L. kitasatonis* in the vagina or urothelium has not yet been described; thus, its identification opens new avenues for research into its potential role as a protective factor or as a marker of microbial balance during pregnancy.

Our findings align with existing evidence on the interplay between the vaginal and urinary microbiomes, where *Lactobacillus* abundance represents a central axis. However, the literature states that in community state types (CSTs) I and II, dominated by *L. crispatus* and *L. gasseri*, respectively, the incidence of UTI is lower, suggesting a protective effect and serving a reference for healthy urobiome. Conversely, CST III and IV, characterized by dominance of *L. iners* and various anerobic bacteria, are associated with higher proportion of UTI outcomes and a dysbiotics state [32]. In contrast, the pattern observed in our study was the opposite of these previously described association. This discrepancy underscores the importance of further investigation to clarify the underlying mechanisms and to explore how population-specific or pregnancy-related factors may shape these microbial dynamics. One possible explanation aligns with the hypothesis proposed by France, who, using the VALENCIA nearest centroid classifier, demonstrated that distinct community state types (CSTs) are differentially represented among women of diverse racial and geographic backgrounds. Specifically, *Lactobacillus*-dominated communities were more prevalent in women of European or Asian descent, whereas non-*Lactobacillus* CSTs were more common among women of African ancestry. These findings highlight how host-associated, environmental, and cultural factors shape the vaginal microbiome and may influence susceptibility to infection and reproductive outcomes [33].

As noted, previously, we were able to distinguish a specific microbial signature for rUTI patients during each trimester. During the first trimester rUTI was characterized by increased abundance of *E. coli*. Additionally, during the second trimester, along with higher abundance of *E. coli*, patients showed lower abundances of *L. kitasatonis* and *L. ultunesis*. Finally, during the third trimester *F. vaginae* (*A. vaginae*) was found solely, in contrast with UTI and control groups. These findings allow the identification of this pattern in the rUTI group, highlighting the importance to create clear cutoffs to determine a better diagnostic measurement, and underscoring the importance of individualized and personalized approaches in clinical practice.

Analysis of alpha diversity revealed contrasting patterns across study groups: in control pregnancies, Evenness and Shannon indices decreased with advancing gestation, whereas in UTI and rUTI patients they progressively increased. This behavior can be explained by species composition: during the first trimester, control samples displayed a higher number of taxa, but as pregnancy advanced, relative diversity was reduced by lactobacilli dominance, reflecting the consolidation of a more stable, uniform microbiome. In contrast, UTI and rUTI patients showed an increase in diversity, suggesting a more unstable ecosystem with multiple taxa competing in a dysbiotic context. These findings are consistent with previous reports indicating that urogenital *Lactobacilli* dominance in normal pregnancies reduces alpha diversity, whereas in infection settings like bacterial vaginosis, loss of this dominance allows expansion of other genera [34,35,36].

In concordance, urotype analysis showed that urotype distribution was not significantly associated with gestational trimester but was significantly different among patient groups (*p* = 0.007364). This indicates that infection status, rather than pregnancy stage, is the primary driver of urinary microbial composition as defined by urotype. In particular, the association between Non *Lactobacillus*-dominated urotypes and infection status underscores their central role as microbial markers of dysbiosis in pregnancy [23,37].

These findings have direct implications for clinical management and diagnostic innovation in pregnancy-associated UTIs. The identification of trimester-specific microbial signatures in rUTI patients supports the development of microbiome-based biomarkers for early detection and risk stratification. Moreover, recognizing that infection status, rather than gestational stage, drives urinary microbial composition highlights the potential for personalized diagnostic and therapeutic strategies targeting microbial imbalance. Integrating microbiome profiling into prenatal care could ultimately improve prevention, diagnosis, and treatment outcomes for both mother and fetus.

**Strengths.** This is the first study to characterize urinary microbiome dynamics during pregnancy in Mexican adolescent women, a clinically and epidemiologically understudied group. Moreover, being a comparative study, the inclusion of three well-defined groups allowed direct comparison of microbial trajectories across different clinical conditions. Furthermore, sequencing identified less-studied taxa such as *L. kitasatonis*, revealing potentially important species-specific patterns. Additionally, this study shows a protective pattern between *L. iners* and a healthy microbiome state, something that was not shown in previous studies, thus being a novel result open for further research regarding variability and biodynamics in different reproductive states, ages, and races. Furthermore, we establish a potential microbial signature for rUTI patients, creating a new route to approach this disease in adolescent pregnant patients.

**Limitations.** Among the limitations of this study, the adolescent cohort was relatively small, limiting statical power to detect subtle associations and precluding robust subgroup analyses. Also, potential confounders such as diet, sexual activity, antibiotic exposure, hygiene, use of douching, and socioeconomic status were not fully controlled, which may influence microbiome composition. Nonetheless, these limitations are counterbalanced by the originality of our results, particularly the description of poorly explored species in this niche and in an adolescent population.

## 4. Conclusions

This study provides the first description of urinary microbiome dynamics during pregnancy in Mexican adolescent patients with and without UTI. Across healthy controls, UTI and rUTI groups, we observed distinct trimester-specific patterns in *Lactobacillus* spp., including progressive increases in *L. iners* and novel detection of *L. kitasatonis* as the dominant taxon. The specific roles of *L. kitasatonis*, *L. helveticus*, and *L. acidophilus* in this context warrant further exploration in larger clinical studies as well as experimental models to clarify their potential protective or disruptive functions in the adolescent urinary microbiome. These findings highlight that not all *Lactobacilillus* spp. confer the same protective effect as a generalized law among all populations and that pregnancy, infection status and adolescent physiology together shape urinary microbial composition.

The observation of microbial profiles and signatures for rUTI underscores the need for personalized approaches to prevent and manage UTI in pregnancy and highlights the need for the establishment of cutoffs for a better clinical approach. Our results reveal population-specific patterns that diverge from reports in adult or non-pregnant cohorts, emphasizing the importance of studying diverse and understudied groups.

Together, these insights expand correct knowledge of the urobiome during adolescent pregnancy and open the opportunity for future research on species-level microbial interactions, host factors, and targeted interventions aimed at improving maternal and neonatal outcomes with a comprehensive intervention.

## 5. Materials and Methods

### 5.1. Inclusion and Exclusion Criteria

Adolescent pregnant patients aged 17 years or younger who attended the Adolescent Clinic of the National Institute of Perinatology (INPer) were considered eligible for this study. Additional inclusion criteria were gestational age between 16 and 24 weeks on their first consultation, absence of urinary tract infection (UTI) at the time of their first consultation, and provision of written informed consent by both the patient and her legal guardian. Patients were excluded if they had received antibiotic treatment within 21 days prior to admission or if they presented with a UTI at their initial consultation. Patients who developed a UTI caused by a non-*E. coli* strain were also excluded. Furthermore, patients were withdrawn from the study if they failed to attend follow-up visits or if they withdrew their assent or informed consent.

### 5.2. Sample Collection and Classification

This was a longitudinal study in which urine samples were collected throughout patient’s pregnancy, with at least one sample obtained during each trimester of gestation until pregnancy resolution. It is important to highlight that all first-trimester samples were obtained from patients with no clinical or laboratory evidence of UTI. Clinical evidence included fever, dysuria, increased urinary frequency, and lower abdominal discomfort. Laboratory evidence was defined as ≥1 × 10^4^ CFU/mL in symptomatic patients, and ≥1 × 10^5^ CFU/mL was considered indicative of infection even in the absence of symptoms. An abnormal urinalysis was also taken into account.

The midstream urine collection technique was used to obtain 86 urine samples from 45 patients treated at the Adolescent Clinic of INPer between January 2018 and December 2019, with at least one sample per trimester of pregnancy.

Each sample was divided into two parts: one for detecting uropathogenic *E. coli* (UPEC) and another for analyzing the urobiome through 16S gene sequencing. Based on the number of UTI identification, patients were categorized into controls (no isolates), single infection (one isolate), and recurrent infection (two or more isolates corresponding to ≥ two acute UTIs within 6 months or at least three within a year).

### 5.3. Sample Processing

Urine samples (5–50 mL) were stored at 4 °C for up to 24 h after collection and at −80 °C for up to two months before processing. DNA extraction was performed using the ZymoBIOMICS™ DNA Microprep Kit (Zimo Research, Cat. D4301; Irvine, CA, USA) with the following protocol modifications: Samples were thawed at room temperature and centrifuged at 3500 rpm for 40 min at room temperature. The supernatant was discarded, and 750 μL of ZymoBIOMICS™ Lysis Solution was added to the pellet, which was then transferred to ZR BashingBead™ Lysis Tubes. Then, the tubes were processed using the MP Biomedicals FastPrep^®^-24 system (MP Biomedicals, Santa Ana, CA, USA) in two cycles at 6.5 m/s for 15 s and three cycles at 4 m/s for 40 s, with a 5 min cooldown between cycles. The manufacturer’s protocol was followed from this point onward.

### 5.4. 16S Sequencing Using Nanopore

16S rRNA gene sequencing was conducted on the MinION Mk1B (Oxford Nanopore Technologies ONT, Oxford, UK) using the 16S Barcoding Kit 1-24 (SQK-16S024; ONT, Oxford, UK) and the “Rapid Amplicon Sequencing—16S Barcoding” protocol with minor modifications. The amplification mix included: 10X Taq Buffer 5 μL, dNTPs (5 mM) 2 μL, MgCl_2_ (25 mM) 3 μL, DNA Taq Polymerase (5 U/μL) 0.7 μL, 16S-barcode primer 10 μL, 100 ng DNA, Nuclease-free water to a final volume of 50 μL.

A total of 100 ng from each library was used, and sequencing was performed in rounds of 10 to 24 samples per round. A total of six rounds were conducted using two R9.4.1 flow cells (ONT, Oxford, UK). Each flow cell was used for three rounds, and the Flow Cell Wash Kit (EXP-WSH004 or EXP-WSH004-XL) (ONT, Oxford, UK) was used according to the manufacturer’s instructions to wash the flow cell between rounds.

Nanopore sequencing using a MinION Mk1B device (ONT, Oxford, UK) was carried out on an Alienware Aurora R11 computer (Dell Inc.; Round Rock, TX, USA), equipped with an NVIDIA RTX 2080 SUPER (8 GB) graphics processing unit (GPU) and an Intel Core i7-11800H (16-core) 11th-generation processor, enabling local base-calling with Guppy.

Sequencing was initiated using MinKNOW version 21.10.4 with high-accuracy base-calling and a quality score (Q score) of ≥ 8, over a period of 5 to 25 h, depending on the amount of data required.

### 5.5. Taxonomic Identification (OTUs)

After sequencing, FASTQ reads were concatenated per sample, trimmed with Nanofilt, and processed using an in-house pipeline with QIIME 2 https://docs.qiime2.org/2024.10/ (accessed on 15 October 2024). The reference database used was SILVA 132 release https://www.arb-silva.de/download/archive/qiime/ (accessed on 15 October 2024).

### 5.6. Statistical Analysis

To assess statistically significant associations in urotype distribution across the different populations, Fisher’s exact test was applied. Differences in microbial abundances among patient groups were evaluated using ANOVA. Alpha diversity analyses were performed in R, employing the Shannon index, evenness, and the number of operational taxonomic units (OTUs) as metrics. Data manipulation and visualization were conducted using the tidyverse, ggplot2, and plotly packages. A comparative analysis of bacterial species’ relative abundances per trimester was performed across three study groups. Average abundances per species and trimester were calculated, and the data for each group were organized into 23-column matrices representing the 23 most abundant species. Data processing and visualization were conducted in Python v3.12.11 using NumPy (array handling and stacking), pandas (DataFrame manipulation), Seaborn v0.13.2 (heatmaps), and Matplotlib v3.10.0 (figures and LaTeX formatting).

## Figures and Tables

**Figure 1 microorganisms-13-02406-f001:**
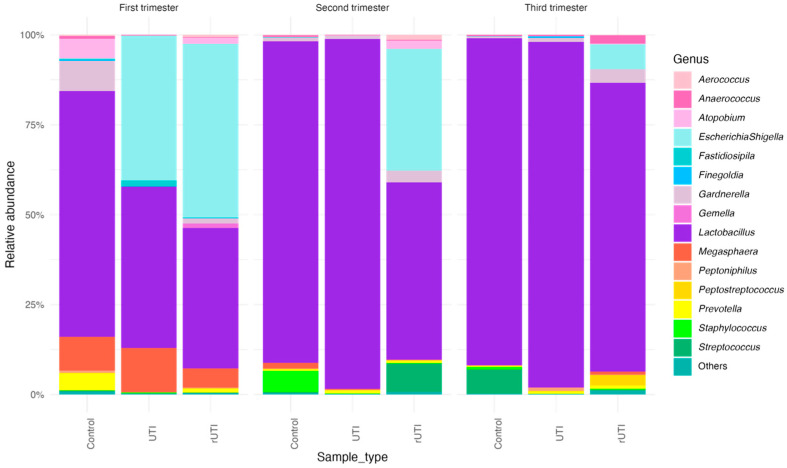
Urinary microbiome profile across pregnancy trimesters at the genus level by cohort, based on 16S rRNA gene sequencing. Stacked bar plots display the sequence abundances of the most abundant bacterial genera in the three study cohorts during pregnancy. The *y*-axis represents the relative abundance expressed as a percentage for each bacterial taxon, while the *x*-axis shows the study participants, grouped by cohort. Each trimester was analyzed independently.

**Figure 2 microorganisms-13-02406-f002:**
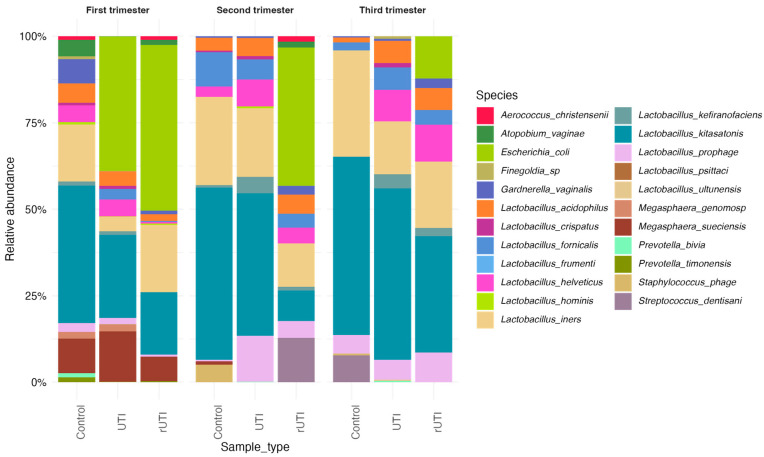
Urinary microbiome profile across pregnancy trimesters at the species level by cohort, based on 16S rRNA gene sequencing. Stacked bar plots display the sequence abundances of the 23 most abundant bacterial species in the three study cohorts during pregnancy. The *y*-axis represents the relative abundance expressed as a percentage for each bacterial taxon, while the *x*-axis shows the study participants, grouped by cohort. Each trimester was analyzed independently.

**Figure 3 microorganisms-13-02406-f003:**
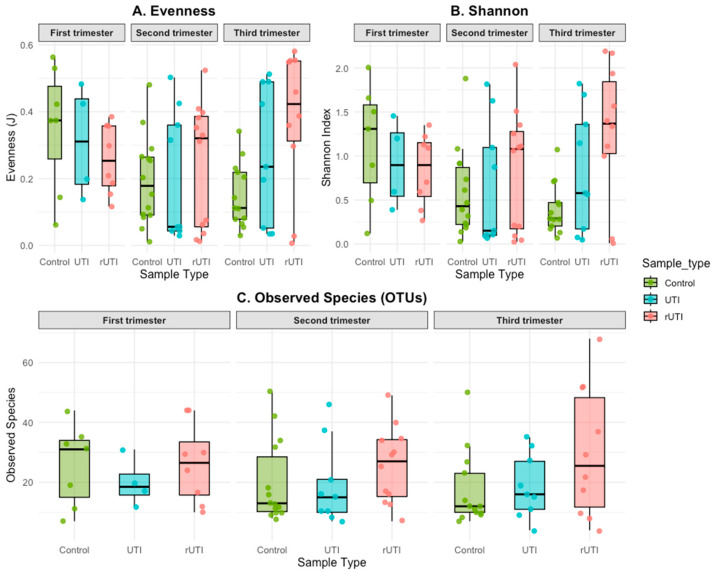
Alpha diversity analysis of the three cohorts separated by trimester based on 16S rRNA gene amplification. Panels show Evenness (**A**), Shannon (**B**), and (**C**) Diversity Observed Species (OTUs).

**Figure 4 microorganisms-13-02406-f004:**
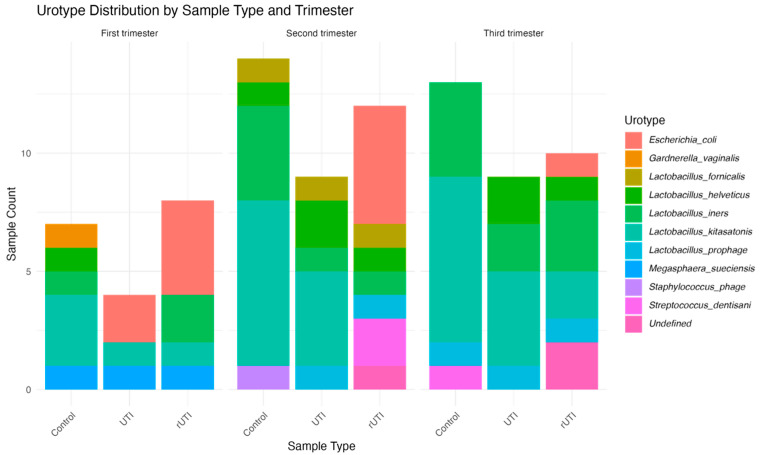
Distribution of urotypes across pregnancy trimesters by cohort, based on the dominant microorganism in each sample. Bar plots show the distribution of the 11 identified urotypes throughout pregnancy within each cohort. The *y*-axis represents the number of samples, and the *x*-axis shows study participants grouped by cohort.

**Figure 5 microorganisms-13-02406-f005:**
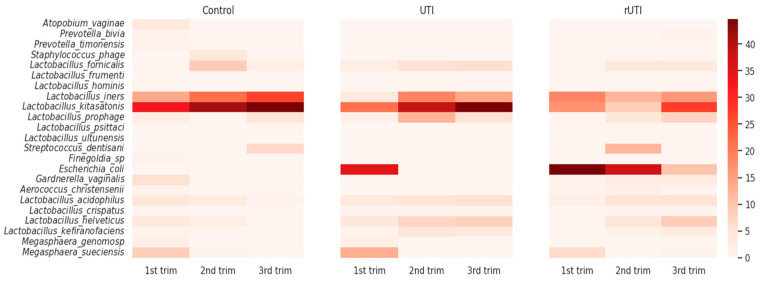
Heatmap showing changes in the abundance of each microorganism throughout pregnancy across the three cohorts. The relative abundance of the most prevalent microorganisms is displayed for each trimester and patient group.

## Data Availability

Data is contained within the article but for more details please contact corresponding author.

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
