# Peer review of "Urobiome Signatures of Recurrent Urinary Tract Infections in Adolescent Pregnancy: A Longitudinal Study"

_microorganisms, 2025, doi:10.3390/microorganisms13102406_

Round 1

Reviewer 1 Report

Comments and Suggestions for Authors

This paper reports the urinary microbiome of pregnant teenagers during the three trimesters of the pregnancy, as well as its association with UTI and recurrent UTI. It provides important information regarding the changes in composition by trimester and study group (healthy, UTI, and recurrent UTI). The paper is well written and the information is clear. The figures are clear, I would only suggest to use a different combination of colors in Fig. 2 so it would be easier to distinguish the different species.

The methods are adequately described, however in section 5.2 the authors mention that the samples were grouped according to the presence of E. coli, whether they found cero, one or 2 or more isolates. But they do not mention if the patients showed UTI symptomatology. Also, authors consider recurrent ITU, but they do not mention if the patients received antibiotic treatment or not, which raises the question, was it a recurrent UTI or could it have been an untreated UTI, especially if there were no symptoms. Since the criterion to determine UTI was the presence of E. coli, could it be possible that samples classifed as healthy have UTI caused by any other etiological agent?

A minor point in the manuscript is that in line 52 it says: with 10% of these cases occurring in from South Asia... I guess it should be: occurring in patients from...

Author Response

REVIEWER 1

  • This paper reports the urinary microbiome of pregnant teenagers during the three trimesters of the pregnancy, as well as its association with UTI and recurrent UTI. It provides important information regarding the changes in composition by trimester and study group (healthy, UTI, and recurrent UTI). The paper is well written and the information is clear. The figures are clear, I would only suggest to use a different combination of colors in Fig. 2 so it would be easier to distinguish the different species.

Authors response: We thank the reviewer for the comments. We changed figure 2 colors as reviewer is right and some of them were ver similar.

  • The methods are adequately described, however in section 5.2 the authors mention that the samples were grouped according to the presence of E. coli, whether they found cero, one or 2 or more isolates. But they do not mention if the patients showed UTI symptomatology. Also, authors consider recurrent ITU, but they do not mention if the patients received antibiotic treatment or not, which raises the question, was it a recurrent UTI or could it have been an untreated UTI, especially if there were no symptoms. Since the criterion to determine UTI was the presence of E. coli, could it be possible that samples classifed as healthy have UTI caused by any other etiological agent?

Authors response: Control patients did not present urinary symptoms. As mentioned in Section 5.1 (lines 430-431), participants were included at their first medical follow-up only if they showed no clinical or laboratory evidence of UTI. To clarify, we added a paragraph in lines 431-434 describing our diagnostic criteria as follows: Clinical evidence included fever, dysuria, increased urinary frequency, and lower abdominal discomfort. Laboratory evidence was defined as ≥1×10⁴ CFU/mL in symptomatic patients, while ≥1×10⁵ CFU/mL was considered indicative of infection even in the absence of symptoms. An abnormal urinalysis was also taken into account.

Treatment was not included as a variable in the analysis, since all patients with evidence of UTI received antibiotic therapy, most commonly with fosfomycin or nitrofurantoin. Some patients also received additional antibiotics for other infections, such as cervicovaginitis, bacterial vaginosis, or bacterial pharyngitis. Therefore, recurrence in some cases might represent reinfection rather than an untreated UTI; however, we were unable to confirm completion of antibiotic treatment in all patients.

As part of the complete project, we analyzed bacterial clonality by pulsed-field gel electrophoresis (data not shown, as this study focuses on microbiome characterization rather than pathogen typing). Results indicated that 65% of recurrent infections were caused by different E. coli strains, whereas 35% were due to the same strain, suggesting bacterial persistence despite antibiotic treatment.

Control patients did not present UTIs caused by E. coli or any other microorganisms. Now this was included in lines 424-425

  • A minor point in the manuscript is that in line 52 it says: with 10% of these cases occurring in from South Asia... I guess it should be: occurring in patients from...

Authors response: Terrible mistake. “.. occurring in patients from …” was corrected.

Reviewer 2 Report

Comments and Suggestions for Authors

The topic of this submitted manuscript is interesting.

Here are some comments, questions, propositions and possible recommendations

line 55-69: not really necessary in this context

line 90-96: a statement is lacking whether all microorgnisms listedare present in every women or only a selection of them in an individual person. Do they persist lifelong in a person are are there fluctuations? The authors should address this problem.

line 1001-107: it has been stated altready above (91-96) that a complex flora exists

line 111: should be mentioned already earlier (line 96)

line 114: no: the above cited literature has also used metagenomic studies

line 238-244: what about intake of probiotics?? (yoghurt)

line 267-274: is the production of lactica acid  lowering the pH (and peroxide) relevant in  urine ???

line 278/79: does this fact plays a role in the urine, where a  normally pH is 5.-65 ?? Is this discussion relevant in this context?

line 300-3: intake of probiotics?

line 319-323: why you have not determined the compositin of the  vaginal flora in parallel?? Is the lactobacilli flora in urine and vagina identical???

line 309: is L. kitasatonis present in the vaginal flora of women in Mexico?

Supplement figur 1 can be deleted, because these data only prove that E.coli is a prevalent cause of urinary tract infections

Author Response

REVIEWER 2

The topic of this submitted manuscript is interesting.

Here are some comments, questions, propositions and possible recommendations

  • line 55-69: not really necessary in this context

Authors response: We thank the reviewer for the valuable comments aimed at improving the manuscript. The revised paragraphs (lines 55–69) are intended to highlight not only the adverse outcomes that may result from adolescent pregnancy itself, but also to raise awareness of the broader social challenges surrounding it. Adolescent pregnancy is often concealed due to fear, stigma, or lack of social support, which leads to delayed access to healthcare services. Such delays substantially increase the risk of adverse outcomes for both mother and newborn, underscoring the importance of early detection, comprehensive prenatal care, and strong social support systems for pregnant adolescents. In this sense, er really want to conserve the paragraph.

  • line 90-96: a statement is lacking whether all microorgnisms listed are present in every women or only a selection of them in an individual person. Do they persist lifelong in a person are are there fluctuations? The authors should address this problem.

Authors response: A new paragraph has been added (lines 100–112) to describe the dynamic fluctuations of the female urinary microbiome (FUM) across the different stages of the female life cycle.

  • line 101-107: it has been stated altready above (91-96) that a complex flora exists

Authors response: Reviewer is right. Paragraph was eliminated.

  • line 111: should be mentioned already earlier (line 96)

Authors response: As we removed one paragraph (previously located at lines 101–107), the paragraph that was originally at line 111 is now directly connected with the preceding section. Consequently, the descriptions of the urobiome in healthy and UTI patients are now presented together (lines 92–122).

  • line 114: no: the above cited literature has also used metagenomic studies

Authors response: The reviewer is correct. Our original statement, “metagenomic studies… are still lacking,” was not intended to suggest that such studies do not exist, but rather that they remain insufficient. To clarify this point, we have revised the sentence to read: “To date, there are only a few metagenomic studies of UTI and recurrent UTI patients.” (line 125).

  • line 238-244: what about intake of probiotics?? (yoghurt)

Authors response: It is already know that probiotic consumtion might modify microbiota, but studies are not conclusive yet.  In fact, information regarding the benefits of consuming probiotics during the reproductive stage to maintain a healthy genitourinary microbiome is still controversial, showing a higher impact on the pregnant and menstrual reproductive systems. Regarding establishing a nutritional regimen, the studies reflect the necessity of taking care of vitamin and mineral plasma levels in women at their reproductive ages to develop a better balance and diversity of the genitourinary tract microbiome, aiming to avoid bacterial vaginosis. High-fiber diets with a balance between animal and vegetable proteins and low in saturated fats with complex carbohydrates (cereals, fruits, and vegetables) provide a beginning for establishing a healthy microbiota in the genitourinary tract. We have a review on urobiome during life cycle of women and we discuss some findings about diet and probiotics use (doi: 10.3390/jcm12124003).

For this particular group we did not consider diet or probiotics use in their interviews. Then, in that sense we decided not to discuss about this particular topic.

  • line 267-274: is the production of lactic acid  lowering the pH (and peroxide) relevant in  urine ???

Authors response: We believe that acidification does not play a crucial role in urine; however, there is currently no direct evidence to support this assertion. It is well established that the urinary microbiota is, to some extent, shared with the vaginal microbiota, which is why some studies refer to it as the urogenital microbiome. In this context, it is extensively documented that Lactobacillus species, by acidifying the vaginal environment, play a key role in maintaining eubiosis. Dysbiotic events in the vagina may therefore influence the urinary microbiome. Additionally, as discussed later, L. iners has the capacity to attach to fibronectin present in the uroepithelium, and this attachment occurs under neutral or only mildly acidic pH conditions, rather than in strongly acidic environments (environment found in urinary tract).

  • line 278/79: does this fact plays a role in the urine, where a  normally pH is 5.-65 ?? Is this discussion relevant in this context?

Authors response: Yes, we consider this relevant because urine is neutral or only mildly acidic; therefore, the presence of L. iners is consistent with a protective role, not through pH reduction, but by acting as a physical and microbial barrier.

  • line 300-3: intake of probiotics?

Authors response: As we previously noted, we do not have information on probiotic intake in our patients. However, it is unlikely that the observed increase in L. acidophilus and L. helveticus is due to probiotics, as this would imply that only the infected patients had consumed them.

  • line 319-323: why you have not determined the compositin of the  vaginal flora in parallel?? Is the lactobacilli flora in urine and vagina identical???

Authors response: We did not determine the composition of the vaginal microbiota, as this was not the aim of the present study and we lacked the financial resources to do so. However, it is well established that the urinary and vaginal niches share microorganisms, and it is likely that the vaginal microbiota serves as a primary source for the urinary microbiota. In fact, we have published a review on the female microbiome in which we compare the composition of the vaginal and urinary microbiotas (doi: 10.3390/jcm12124003).

  • line 309: is L. kitasatonis present in the vaginal flora of women in Mexico?

Authors response: We do not currently have this information, but we hope to investigate it in the near future. A study on the vaginal microbiota of Mexican pregnant women (2011) used PCR targeting the most commonly reported Lactobacillus species, and therefore did not include L. kitasatonis (doi: 10.1155/2011/851485). A more recent study (2022) in 10 pregnant Mexican patients reported Lactobacillus as the dominant taxa; however, species-level data were not provided (doi: 10.1007/s00284-022-02918-1). Finally, a 2025 study examining vaginal microbiota associated with yeast infection also reported results only at the genus level (doi: 10.3390/biotech14020031).

  • Supplement figure 1 can be deleted, because these data only prove that coli is a prevalent cause of urinary tract infections.

Authors response: Figure S1 not only confirms that E. coli is a prevalent cause of urinary tract infection, but also shows that statistically significant differences were observed only between rUTI and control patients (lines 174–175). As described in the Methods section, patients were included only if they had no clinical or laboratory evidence of UTI. Therefore, we propose that, in cases of recurrence, some E. coli strains may persist in the host asymptomatically. This persistence could serve as a potential biomarker for this type of infection in patients without overt UTI.

Reviewer 3 Report

Comments and Suggestions for Authors

See the attachment.

Author Response

REVIEWER 3

Dear Authors,

Below are my comments on the manuscript, which will help the authors improve the text by clarifying and expanding on the issues raised.

  • The title of the article accurately reflects the content of the publication. Still, I would consider changing the word “signatures” to “profiles” or “patterns,” as “signatures” quite clearly indicates specific indicators or markers or species of microorganisms, which may not be the best term in the case of the urinary tract microbiome, which changes dynamically during pregnancy.

Authors response: Reviewer is quite right as we described the microbiota pattern in our 3 patients groups. Any way, our study was able to distinguish a species signature only in rUTI patients group. In that sense, we discussed about the importance of taking this results to continue reseach on diagnosis field in order to detect the posibility of rUTI presentation in this vulnerable population even before it is clinically or laboratory evident (324-332). The early diagnosis of UTI, and specially rUTI in pregnant adolescents might contribute in diminishing the adverse outcomes derived from urinary infections.

  • The abstract of the article adequately reflects the content of the research, its analysis, results, and conclusions. It is worth considering enriching the abstract with the size of the study groups and adding information about the sequencing method – Oxford nanopore.

Authors response: we believe  the reviewer is right and we included the number of samples and the sequencing method in the abstract.

  • In the abstract and elsewhere in the article, please use the current name Atopobium vaginae, namely Fannyhessea vaginae, in accordance with the reference: Nouioui I, Carro L, García- López M, Meier-Kolthoff JP, Woyke T, Kyrpides NC, Pukall R, Klenk HP, Goodfellow M, Göker M. Genome-Based Taxonomic Classification of the Phylum Actinobacteria. Front Microbiol. 2018 Aug 22;9:2007. doi: 10.3389/fmicb.2018.02007. PMID: 30186281; PMCID: PMC6113628. I suggest that the name Atopobium vaginae be given in parentheses, due to the relatively widespread use of this name.

Authors response: We thank the reviewer for this observation, which is correct. We have decided to use both names throughout the text, as A. vaginae is commonly used in the literature. Additionally, the database employed for sequence analysis uses Atopobium vaginae as the species name.

  • In the introduction, the authors provide a comprehensive overview of urinary tract infections in young pregnant women, including their possible causes and epidemiological and social aspects. Some parts of the introduction are somewhat redundant (e.g., lines 79-87 and 97-113). Following their reduction, I suggest paying more attention to the ethno-geographical aspect of the epidemiology of urinary tract infections and the microbiome of the reproductive tract, see reference: France MT, Ma B, Gajer P, Brown S, Humphrys MS, Holm JB, Waetjen LE, Brotman RM, Ravel J. VALENCIA: a nearest centroid classification method for vaginal microbial communities based on composition. Microbiome. 2020 Nov 23;8(1):166. doi: 10.1186/s40168-020-00934-6. PMID: 33228810; PMCID: PMC7684964.

Authors response: The reviewer is correct that some parts of the Introduction were redundant. We have removed one paragraph and revised the wording in the description of the microbiome in healthy individuals and during infection. The Introduction does not mention vaginal microbial communities; however, this topic is addressed in the Discussion (lines 335–344). Regarding ethnicity, our study population consists entirely of adolescent patients in Mexico, corresponding to a similar or homogeneous ethnic background. Nevertheless, we have now included a discussion of the potential influence of ethnicity on the microbiome in the Discussion section (lines 355–363).

  • The results are presented clearly and correctly described, and are adequately illustrated with figures.

Authors response: We thank the reviewer.

  • The text from the manuscript template present in lines 124-126 should be removed.

Author response: Done

  • Please write Latin names in italics, e.g., lines 137, 140, 141, 144, and elsewhere in the manuscript.

Author response: We are very sorry, we revised the manuscript to have all the microorganisms names in italics.

  • A key omission in this section is the lack of a description of beta-diversity. The absence of beta- diversity analysis limits the ability to assess the overall ecological differences and the impact of clinical factors on microbiome structure.

Authors response: We initially decided not to include beta diversity because no statistically significant differences were observed between groups. However, the reviewer is correct that it is important to present these results. Therefore, we have added a new supplementary figure (Figure S4) showing the beta diversity analyses, which are now also referenced in the Results section (lines 218–227).

  • The discussion is conducted correctly, referring to a large number of appropriately selected literature sources. A very interesting aspect is the observation and description of kitasatonis and L. iners, although in my opinion, this discussion should be broader and also cover the roles of these species in the vaginal microbiota of both pregnant and non-pregnant women.

Authors response: The Discussion already includes a section describing L. iners (lines 284–320). We have further enriched this section by adding relevant findings on the vaginal microbiota (lines 307–311). In contrast, L. kitasatonis has not been previously reported in humans with respect to its function. We have now included a new paragraph summarizing interesting findings that may support its potential protective role (lines 337–344).

  • In my opinion, the discussion lacks a clear definition of the potential possibilities for translating the research results into clinical practice.

Author response: The reviewer is correct. Although we attempted to explain the clinical implications of the study, we recognized that this was not presented explicitly. We have therefore added a paragraph addressing this at the end of the Discussion section (lines 390–397).

  • The materials and methods section discusses patient eligibility in detail. However, I would ask for a correction or clarification of “recurrent infection (two or more isolates)” as it can be understood as the cultivation of >2 bacterial strains from a single sample, while rUTI is defined as “≥ two acute UTIs within 6 months or at least three within a year” https:// ncbi.nlm.nih.gov/books/NBK557479/.

Authors response: The reviewer is right we only asume isolations but not time. We have modified the rUti classification (lines 411-414).